# Anti-Inflammatory Activity of Four Triterpenoids Isolated from *Poriae Cutis*

**DOI:** 10.3390/foods10123155

**Published:** 2021-12-20

**Authors:** Lijia Zhang, Mengzhou Yin, Xi Feng, Salam A. Ibrahim, Ying Liu, Wen Huang

**Affiliations:** 1College of Food Science and Technology, Huazhong Agricultural University, Wuhan 430070, China; 13297021290@163.com (L.Z.); mz324666@163.com (M.Y.); yingliu@mail.hzau.edu.cn (Y.L.); 2Department of Nutrition, Food Science and Packaging, San Jose State University, San Jose, CA 95192, USA; xi.feng@sjsu.edu; 3Department of Family and Consumer Sciences, North Carolina Agricultural and Technical State University, 171 Carver Hall, Greensboro, NC 27411, USA; ibrah001@ncat.edu

**Keywords:** *Poriae Cutis*, triterpenoid, purification, identification, anti-inflammatory

## Abstract

In this study, triterpenoid compounds from *Poriae Cutis* were separated by high-speed countercurrent chromatography (HSCCC) and identified using ultra-high performance liquid chromatography quadrupole time-of-flight tandem mass spectrometry (UHPLC-QTOF-MS/MS) and nuclear magnetic resonance (NMR). The in vitro anti-inflammatory activities of the purified triterpenoids on RAW 264.7 cells were also investigated. Triterpenoids, poricoic acid B, poricoic acid A, dehydrotrametenolic acid, and dehydroeburicoic acid were obtained; their levels of purity were 90%, 92%, 93%, and 96%, respectively. The results indicated that poricoic acid B had higher anti-inflammatory activity than those of poricoic acid A by inhibiting the generation of NO in lipopolysaccharide (LPS)-induced RAW 264.7 cells. However, dehydrotrametenolic acid and dehydroeburicoic acid had no anti-inflammatory activity. In addition, the production of cytokines (TNF-α, IL-1β, and IL-6) in cells treated with poricoic acid B decreased in a dose-dependent manner in the concentration range from 10 to 40 μg/mL. The results provide evidence for the use of *Poriae Cutis* as a natural anti-inflammatory agent in medicines and functional foods.

## 1. Introduction

*Poria cocos* (Schw.) Wolf (“fuling” in Chinese) is a widely used medicinal fungus. According to the Chinese Pharmacopoeia, about one-tenth of traditional Chinese medicine preparations include *P. cocos* [1]. Additionally, *P. cocos* is used as a dietary supplement and in foods such as soups, dishes, snacks, and desserts for its health-promoting benefits. It can be divided into four parts, namely, *Poriae Cutis* (the epidermis, “fulingpi” in Chinese), *Rubra Poria* (the pink part near the epidermis), *White Poria* (the middle part, “baifuling” in Chinese), and *Poria cum Radix Pini* (the middle-plus-inner part, “fushen” in Chinese) [2]. Different parts of *Poria cocos* have different pharmacological effects and clinical applications. Both *P. cocos* and *Poriae Cutis* are officially included in the Chinese Pharmacopoeia. Clearing damp and promoting diuresis are traditional Chinese medicinal uses for *Poria cocos*. *Poriae Cutis* is mainly used to promote urination in clinics. Nowadays, *White Poria* is widely used due to its diuretic, tonic, and sedative functions. However, *Poriae Cutis* is treated as agricultural waste during the harvest periods and is rarely used [1,3].

The functional constituents in *P. cocos* mainly contain triterpenes and polysaccharides, while the main bioactive constituent of *Poriae Cutis* is triterpenoid. Triterpenes isolated from *P. cocos* can be divided into four subgroups based on their chemical structure: lanosta-8-ene type, lanosta-7,9(11)-diene type, 3,4-seco-lanostan-8-ene type, and 3,4-seco-lanostan-7,9(11)-diene type compounds [4]. In prior research, 59 types of triterpenoids were separated from *P. cocos*; 41 types were isolated from *Poriae Cutis* [2,4]. The total content of triterpenes and bioactive triterpenes in *Poriae Cutis* was found to be higher than in the sclerotia of *P. cocos* [1,5,6,7], and triterpenoids from the sclerotia of *P. cocos* showed anti-inflammatory activity [8,9]. It was reported that seco-lanostane triterpenoid come from the sclerotia of *P. cocos* and could be a promising lead compound for the development of anti-inflammatory agents for use in pharmaceuticals and functional foods [8]. Triterpenes have attracted attention due to their anti-inflammatory [8,10,11], anticancer [12,13,14], and diuretic properties [3,15]. Of the many reports on the biological activity of triterpenes in *Poriae Cutis*, most have focused on antitumor and diuretic activities. However, there are few studies on the anti-inflammatory activity of triterpenoids in *Poriae Cutis*.

The objectives of this study were to separate triterpenoids from *Poriae Cutis* by HSCCC chromatography, and then identify them with UHPLC-QTOF-MS/MS and NMR spectra. The anti-inflammatory activity of triterpenoids was also determined using RAW 264.7 mouse macrophage cells.

## 2. Materials and Methods

### 2.1. Crude Sample Preparation

*Poria cocos* was obtained from the local market in Lou tian (Hubei Province, China). The powder of *Poriae Cutis* (50 g) was extracted with n-butanol-water (1:1, *v*/*v*) solution (2 L) at 60 °C under reflux for 40 min. After filtration, the filtrate was concentrated by a rotary evaporator. Concentrated filtrate was suspended in distilled water and extracted with ethyl acetate three times. The combined ethyl acetate solution was evaporated to dryness to yield crude triterpenes from *Poriae Cutis*, which were used for subsequent HSCCC separation [16].

### 2.2. HSCCC Separation

A high-speed counter-current chromatography (HSCCC) with a two-phase solvent system (TBE-300C, Shanghai Tauto Biotech Co., Ltd., Shanghai, China) with n-hexane-ethyl acetate-MeOH-water (3:6:4:2, *v*/*v*/*v*/*v*) was used; the stationary phase was the upper phase, and the mobile phase was the lower phase. Crude triterpene (100 mg) was dissolved in the mobile phase (10 mL). After two-phase ultrasonic degassed for 15 min, the sample solution was injected through the injection valve when it reached hydrodynamic equilibrium. Following the stationary phase, the column was filled entirely at a flow rate of 30 mL/min; the lower phase was pumped at a flow rate of 3 mL/min. The rotation was set at 800 rpm with a detection wavelength of 242 nm. Six peaks (P-1, P-2, P-3, P-4, P-5, and P-6) were separately collected and evaporated to dryness to yield purified fractions [16].

### 2.3. HPLC Analysis

The six purified fractions (P-1, P-2, P-3, P-4, P-5, and P-6) were dissolved in methanol and then filtered by a 0.45 µm syringe filter for HPLC analysis. HPLC (e2695, Waters Corporation, Milford, MA, USA) analyses were performed with an InertSustainTMAQ-C18 column (4.6 mm × 250 mm, 5 µm, Shimadzu, Shanghai, China) at 25 °C. The mobile phase was composed of solvent A (0.05% (*v*/*v*) phosphoric acid) and solvent B (methanol). The gradient mode was as follows: 0–15 min, 30–25% A, 70–75% B; 15–25 min, 25–21% A, 75–79% B; 25–40 min, 21–10% A, 79–90% B; 40–65 min, 10% A, 90% B. The injection volume was 10 µL and the flow rate was 1.0 mL/min with a detection wavelength of 242 nm.

### 2.4. UHPLC-QTOF-MS/MS Analysis

UHPLC (1100-VL, Waters Corporation, Milford, MA, USA) analyses were performed with an ACQUITY UPLC^®^ BEH C18 column (2.1 mm × 100 mm, 1.7 µm, Waters Corporation, Milford, MA, USA) at 30 °C. The mobile phase was composed of solvent A (water) and solvent B (methanol). The gradient mode was as follows: 0–7 min, 44–23% A, 56–77% B; 7–14 min, 23–15% A, 77–85% B; 14–24 min, 15% A, 85% B. The injection volume was 1 µL and the flow rate was kept at 0.3 mL/min. Mass spectrometry conditions were set with the ESI source in negative ionization mode, drying gas temperature at 350 °C, drying gas flow 600 L/h, and capillary voltage 2.4 kV. The full scan range of *m/z* 10–1000 was performed for mass detection.

### 2.5. NMR Spectra Analysis

Samples were dissolved in methanol-d4 and pyridine-d5 for NMR spectra measurement. NMR spectra analyses were performed by an AVANCE III 400 MHz NMR spectrometer (Bruker Corporation, Zurich, Switzerland). ^1^H-NMR and ^13^C-NMR data were analyzed and processed by Meatrenove 14.0.

### 2.6. Anti-Inflammatory Activity of Triterpenoids from Poriae Cutis In Vitro

#### 2.6.1. Cell Viability Assay

RAW 264.7 cells (acquired from Chinese Academy of Sciences) were maintained in Dulbecco’s modified Eagle’s medium (DMEM, Gibco Company, New York, NY, USA), supplemented with fetal bovine serum (FBS) (10%, *v*/*v*) (Gibco Company, USA) and antibiotics (1%, *v*/*v*) (penicillin (100 U/mL) and streptomycin (100 μg/mL)) in a humidified incubator at 37 °C with 5% CO_2_. Cells were plated into 96-well plates at 1 × 10^5^ cells/mL. After incubating for 24 h, the RAW 264.7 cells were added to 100 μL of four triterpenoids from *Poriae Cutis* with different concentrations (poricoic acid A and poricoic acid B: 10, 20, 30, 40, and 50 μg/mL; dehydrotrametenolic acid and dehydroeburicoic acid: 0.125, 0.25, 0.5, 1, and 5 μg/mL), DMEM basic medium (control group) or lipopolysaccharide (LPS) (Sigma Company, St. Louis, MO, USA) (1 μg/mL, positive control) for 24 h. Next, the growth medium was removed and 100 μL of 3-(4,5-dimethylthiazole-2)-2,5-diphenyltetrazolium bromine salt (MTT) (American Biosharp Company, Houston, TX, USA) solution (0.5 mg/mL) was added for 4 h. Subsequently, the conditioned medium was eliminated and DMSO (150 μL) was added, and cell viability was detected using optical density (OD) at 490 nm using a Multiskan GO automatic microplate reader (Thermo, Waltham, MA, USA) [17].

#### 2.6.2. NO Determination

RAW 264.7 cells were plated into 24-well plates (8 × 10^5^ cells/mL) and cultured as described above. The cells were then simultaneously induced with LPS (1 μg/mL) and various doses of the four triterpenoids from *Poriae Cutis* (poricoic acid A and poricoic acid B, 10, 20, 30, and 40 μg/mL; dehydrotrametenolic acid and dehydroeburicoic acid, 0.125, 0.25, 0.5, and 1 μg/mL), DMEM basic medium (control group), or LPS (1 μg/mL, positive control) for 24 h. Next, the conditioned medium (100 μL) was mixed with Griess reagent (100 μL) in each well and incubated for 10 min. Finally, the absorbance at 540 nm was measured using a Multiskan GO automatic microplate reader. NaNO_2_ was used as a standard to calculate the released NO concentration [18].

#### 2.6.3. Cytokines Assays of TNF-α, IL-1β, and IL-6

RAW 264.7 cells were cultured in 24-well plates (8 × 10^5^ cells/mL), to which LPS (1 μg/mL) was added and various doses of poricoic acid B (10, 20 and 40 μg/mL), DMEM basic medium (control group), or LPS (1 μg/mL, positive control) for 24 h. The cells were then centrifuge at 2400 r/min for 5 min and the supernatant was collected. The secretions of cytokines in the supernatant were tested using a TNF-α, IL-1β, and IL-6 ELISA kit (Guangzhou Xinbosheng Biotechnology Co., Ltd., Guangzhou, China).

### 2.7. Data Analysis

The structural formula of the compound was drawn by ChemDraw 16.0. Statistical differences between treatments were determined using ANOVA and Duncan’s or Tamhane’s test at a significance level of 0.05 or 0.01.

## 3. Results and Discussion

### 3.1. HSCCC Separation

HSCCC is widely utilized in the isolating of natural products. It has high stability, good reproducibility, the non-requirement of a solid carrier, a low risk of nonreversible adsorption, and a high sample recovery rate [19,20]. Fractions of *P. cocos* were often obtained by means of a silica gel column chromatography and eluted successively [21], but this requires a complicated operation process and causes sample loss. It was reported that HSCCC is an effective methodology for isolating of high-purity triterpenoid compounds in *Inonotus obliquus* [20]. In the present study, HSCCC was used to separate triterpenoid compounds from *Poriae Cutis*. As shown in Figure 1, compounds P-1 (14.37%), P-2 (7.93%), P-3 (20.38%), P-4 (35.64%), P-5 (19.07%), and P-6 (2.59%) were efficiently separated within 360 min with a 50% retention rate. After elution, all of the fractions were dried and the collected fractions were identified by HPLC.

### 3.2. HPLC Analysis

Among the six fractions collected through HSCCC, only P-3, P-4, P-5, and P-6 had a purity of higher than 90% (90%, 92%, 93%, and 96%, respectively). Therefore, the follow-up experiments focused on these four fractions, and we named them compound-**1**, compound-**2**, compound-**3**, and compound-**4**, respectively. A total of 5.20 mg of compound-**1**, 20.50 mg of compound-**2**, 0.4 mg of compound-**3**, and 0.1 mg of compound-**4** were separated from the above sample (100 mg) by HSCCC. As shown in Figure 2, the retention time of the four fractions was 33.23 min (compound-**1**), 36.20 min (compound-**2**), 55.45 min (compound-**3**), and 59.88 min (compound-**4**).

### 3.3. Identification of Compounds

The structures of all compounds were characterized by ESI–MS, ^1^H-NMR, and ^13^C-NMR spectra and are shown in Appendix A. Compound-1 (Appendix A): ESI-MS (*m/z*): 483.3104 [M-H]^-^. UV λ_max_: 242 nm. The ^1^H- and ^13^C-NMR spectra revealed H-11 (δ 5.21)/C-11 (δ 120.7), H-7 (δ 5.17)/C-7 (δ 119.0), C-9 (δ 138.3), C-8 (δ 142.3), H-18 (δ 0.99), H-19 (δ 0.93), H-26 (δ 1.50), H-27 (δ 1.58), H-29 (δ 1.68), H-30 (δ 1.50), H-16 (δ 4.54), H-28 (δ 4.67, 4.76), H-24 (δ 5.27), C-3 (δ 178.3), C-21 (δ 180.0). Based on the structural features of triterpene compounds and the data given in the literature, compound-**1** was identified as poricoic acid B [1,17,20,22,23,24]. Poricoic acid B belongs to the 3,4-seco-lanostan-7,9(11)-diene type triterpenes [4]. Its chemical structure is shown in Figure 3.

Compound-**2** (Appendix A): ESI-MS (*m/z*): 497.3266 [M-H]^-^. UV λ_max_: 242 nm. The ^1^H- and ^13^C-NMR spectra presented H-11 (δ 5.27)/C-11 (δ 119.5), H-7 (δ 5.21)/C-7 (δ 117.8), C-9 (δ 137.1), C-8 (δ 141.1), H-18 (δ 1.09), H-19 (δ 1.02), H-26 (δ 0.98), H-27 (δ 0.99), H-29 (δ 1.74), H-30 (δ 1.50), H-16 (δ 4.54), H-28 (δ 4.77, 4.82), C-3 (δ 176.9), C-21 (δ 178.5), C-24 (δ 155.3), C-25 (δ 33.6), C-31 (δ 106.7). Based on the structural features of triterpene compounds and the data given in the literature, compound-**4** was identified as poricoic acid A [1,16,19,21,22]. Poricoic acid A belongs to the 3,4-seco-lanostan-7,9(1 1)-diene type triterpenes [4]. Its chemical structure is shown in Figure 3.

Compound-**3** (Appendix A): ESI-MS (*m/z*): 453.3370 [M-H]^-^. UV λ_max_: 243 nm. The ^1^H- and ^13^C-NMR spectra indicated that H-11 (δ 5.37)/C-11 (δ 116.5), H-7 (δ 5.61)/C-7 (δ 121.0), C-9 (δ 146.4), C-8 (δ 142.7), H-18 (δ 1.01), H-19 (δ 1.07), H-26 (δ 1.67), H-27 (δ 1.62), H-28 (δ 1.22), H-29 (δ 1.13), H-30 (δ 1.06), H-3 (δ 3.46), H-24 (δ 5.33), C-3 (δ 77.9), C-21 (δ 178.9), C-24 (δ 124.8), C-25 (δ 131.5). Based on the structural features of triterpene compounds and the data given in the literature, compound-**4** was identified as dehydrotrametenolic acid [1,2,19]. Dehydrotrametenolic acid belongs to the lanosta-7,9(11)-diene type triterpenes [4]. Its chemical structure is shown in Figure 3.

Compound-**4** (Appendix A): ESI-MS (*m/z*): 467.3518 [M-H]^-^. UV λ_max_: 243 nm. The ^1^H- and ^13^C-NMR spectra showed that H-11 (δ 5.62)/C-11 (δ 116.5), H-7 (δ 5.61)/C-7 (δ 121.0), C-9 (δ 146.4), C-8 (δ 142.7), H-18 (δ 1.02), H-19 (δ 1.08), H-26 (δ 1.04), H-27 (δ 1.05), H-28 (δ 1.23), H-29 (δ 1.14), H-30 (δ 1.08), H-3 (δ 3.46), H-31 (δ 4.90, 4.95), C-3 (δ 77.9), C-21 (δ 178.9), C-24 (δ 160.8), C-31 (δ 106.8). Based on the structural features of triterpene compounds and the data given in the literature, compound-**4** was identified as dehydroeburicoic acid [1,24]. Dehydroeburicoic acid belongs to the lanosta-7,9(11)-diene type triterpenes [4]. Its chemical structure is shown in Figure 3.

### 3.4. In Vitro Anti-Inflammatory Activity of Triterpenoids

#### 3.4.1. Effects of Triterpenoids on Viability of RAW264.7 Cells

The cytotoxic effect of poricoic acid A, poricoic acid B, dehydrotrametenolic acid, and dehydroeburicoic acid on RAW 264.7 cells was evaluated by an MTT assay. It is generally acknowledged that when adding samples, if the cell viability is below 90%, it indicates cytotoxicity. If the cell viability is higher than this critical value, the sample is non-cytotoxic and can be used for further research [25]. The viabilities of the RAW 264.7 cells of the four triterpenoids from *Poriae Cutis* were different and are shown in Figure 4. As shown in Figure 4A, the two treatment groups of poricoic acid A and poricoic acid B at concentrations of 10–30 μg/mL and 10–40 μg/mL had no cytotoxicity on macrophages. As shown in Figure 4B, the non-cytotoxic concentrations of dehydrotrametenolic acid and dehydroeburicoic acid were 0.125–1 and 0.125–0.5 μg/mL, respectively. However, Lee et al. [8] reported a the cell viability of >90% for n-butanol fraction concentrations of up to 50 μg/mL. This could be due to the differences in the sample purities.

In summary, the maximum concentrations of poricoic acid A, poricoic acid B, dehydrotrametenolic acid, and dehydroeburicoic acid without significant cell activity inhibition on RAW264.7 cells were 30, 40, 1, and 0.5 μg/mL, respectively. The difference may be related to their chemical structures. Poricoic acid A and poricoic acid B belong to 3,4-seco-lanostan-7,9(11)-diene type triterpenes, which were less cytotoxic to RAW264.7 cells; dehydrotrametenolic acid and dehydroeburicoic acid are both lanosta-7,9(11)-diene type triterpenes, which were reported to have a higher cytotoxicity [4]. Dong et al. [26] also reported that seco-lanostane triterpenoid from the epidermis of *Poria cocos* showed weak cytotoxic activities on MGC-803 and HepG2 cell lines.

#### 3.4.2. Effects of Triterpenoids on NO Production of RAW 264.7 Cells

Nitric oxide (NO) has been acknowledged as an important signaling molecule related to inflammatory responses [24]. Natural compounds were reported to affect the secretion of NO, which helped to develop drugs to treat inflammatory diseases [27,28]. LPS is an important proinflammatory factors, which can increase the production of NO [29]. The inhibitory effects of four triterpenoids from *Poriae Cutis* on NO production in LPS-induced RAW 264.7 cells in co-culture experiments are shown in Figure 5. The NO production level of macrophages in the control group was weak, but the NO production level increased significantly after LPS stimulation, which indicated that RAW264.7 cells had already developed an inflammatory state under the induction of LPS. As shown in Figure 5A, the RAW 264.7 cells treated with either poricoic acid A at concentrations of 10–20 μg/mL or poricoic acid B at concentrations of 10 μg/mL showed no change in the secretions of NO (*p* > 0.05), but the change became significant with a further increase in concentration. The NO production in cells treated with poricoic acid A significantly decreased to 47% at 30 μg/mL. Poricoic acid B dose-dependently decreased NO production at concentrations of 20, 30, and 40 μg/mL, and considerably suppressed NO production to 29%, 34%, and 68%, respectively. However, we found that the IC_50_ of poricoic acid A that inhibited NO production in LPS-activated cells was 18.12 μΜ (8.77 μg/mL) [8]. As shown in Figure 5B, dehydrotrametenolic acid and dehydroeburicoic acid had no effect on the secretions of NO and they seemed to have no anti-inflammatory activity. The results are consistent with those in a previous report [9]. Additionally consistent with the report is that the inhibition of NO of seco-lanostane triterpenoid was higher than that of lanostane triterpenoid [8].

#### 3.4.3. Effects of Poricoic Acid B on Cytokines Production of RAW 264.7 Cells

The anti-inflammatory activities of four triterpenes (poricoic acid A, poricoic acid B, dehydrotrametenolic acid, and dehydroeburicoic acid) were assessed by an LPS-stimulated NO production model on RAW 264.7 cells. The results revealed that poricoic acid B exhibited a strong NO production inhibitory activity in a dose-dependent manner. Thus, the secretion of cytokines including tumor necrosis factor (TNF)- α, interleukin (IL)-1β, and interleukin (IL)-6 was measured with poricoic acid B in a medium of RAW 264.7 in co-culture experiments, which were quantified by an ELISA assay.

It has been reported that LPS-induced RAW 264.7 cells secrete pro-inflammatory factors TNF-α, IL-1β, and IL-6 during the inflammation process [30,31]. As shown in Figure 6, the production of TNF-α, IL-1β, and IL-6 by cells treated with poricoic acid B showed a dose-dependent decrease in the concentration range from 10 to 40 μg/mL in the co-culture experiments, all of which were more than in the control group (193 pg/mL), but less than in the LPS-treated group (2.63 × 10^5^ pg/mL). The result indicated that LPS increased the release of pro-inflammatory cytokines [28], and poricoic acid B reduced the secretion of cytokines. TNF-α secretion from the poricoic acid B treated group decreased significantly in the concentration range of 10–40 μg/mL. The value was the lowest at a concentration of 20 μg/mL, which was reduced by 32% compared to that in the LPS-treated group (Figure 6A). The poricoic acid B treated group did not show a significant difference in IL-1β at relatively lower concentrations (10 μg/mL) but displayed significant decreases at higher concentrations. At the doses of 20 and 40 μg/mL, the secretion of IL-1β decreased by 41% and 77% in comparison to the LPS-treated group (Figure 6B), respectively. With the increase in the concentration of poricoic acid B, the amount of IL-6 production gradually decreased. Compared to the LPS-treated group, the secretion of IL-6 decreased by 30%, 53%, and 66% under the poricoic acid B treatment concentrations of 10, 20, and 40 μg/mL, respectively (Figure 6C). 3,4-seco-dammarane triterpenoid saponins isolated from the leaves of *Cyclocarya paliurus* showed similar inhibition of IL-6 and TNF-α [32].

## 4. Conclusions

In this study, four triterpenoids were isolated from *Poriae Cutis*: two 3,4-seco-lanostan-7,9(11)-diene type triterpenes (poricoic acid B and poricoic acid A) and two lanosta-7,9(11)-diene type triterpenes (dehydrotrametenolic acid and dehydroeburicoic acid). The cytotoxicity of the 3,4-seco-lanostan-7,9(11)-diene type triterpenes was lower than that of the lanosta-7,9(11)-diene type triterpenes. The 3,4-seco-lanostan-7,9(11)-diene type triterpenes were found to inhibit the NO generation in LPS-induced RAW264.7 cells. However, the lanosta-7,9(11)-diene type triterpenes had no effect on the production of NO. In addition, poricoic acid B significantly inhibited the secretion of TNF-α, IL-1β, and IL-6 in LPS-induced RAW 264.7 cells. Our results indicated that 3,4-seco-lanostan-7,9(11)-diene type triterpenes exhibited more anti-inflammatory activities than lanosta-7,9(11)-diene type triterpenes, and provided experimental evidence that *Poriae Cutis* is a potential source of natural anti-inflammatory agents for use in pharmaceuticals and functional foods.

## Figures and Tables

**Figure 1 foods-10-03155-f001:**
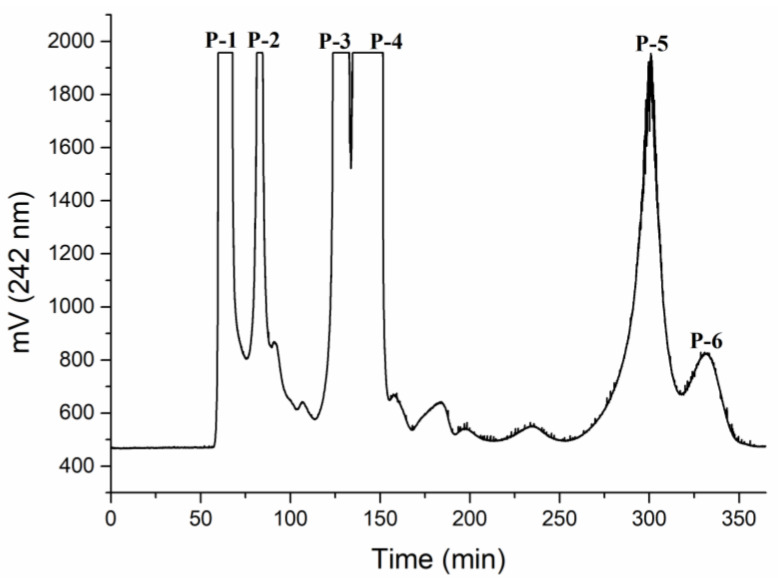
Elution profile of triterpenoids by HSCCC chromatogram. n-hexane-ethyl acetate-MeOH-water (3:6:4:2, *v*/*v*/*v*/*v*); flow rate: 3 mL/min; detection wavelength: 242 nm; rotational speed: 800 rpm; sample size: 100 mg.

**Figure 2 foods-10-03155-f002:**
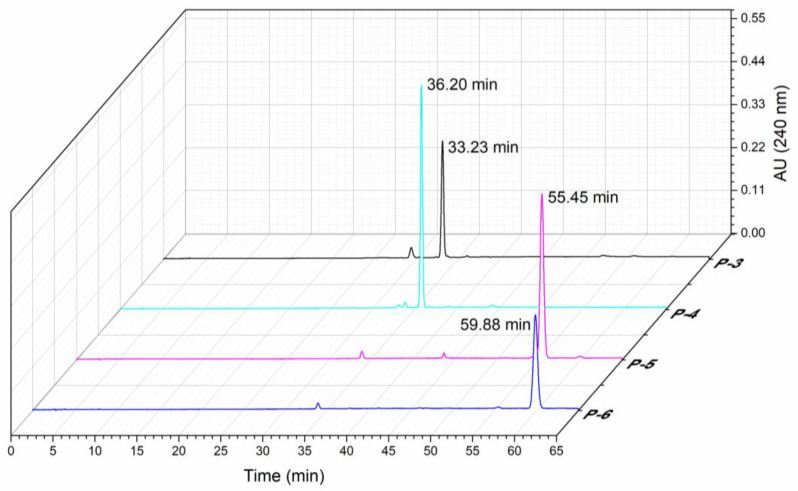
HPLC chromatogram of four kinds of triterpenoids obtained by HSCCC.

**Figure 3 foods-10-03155-f003:**
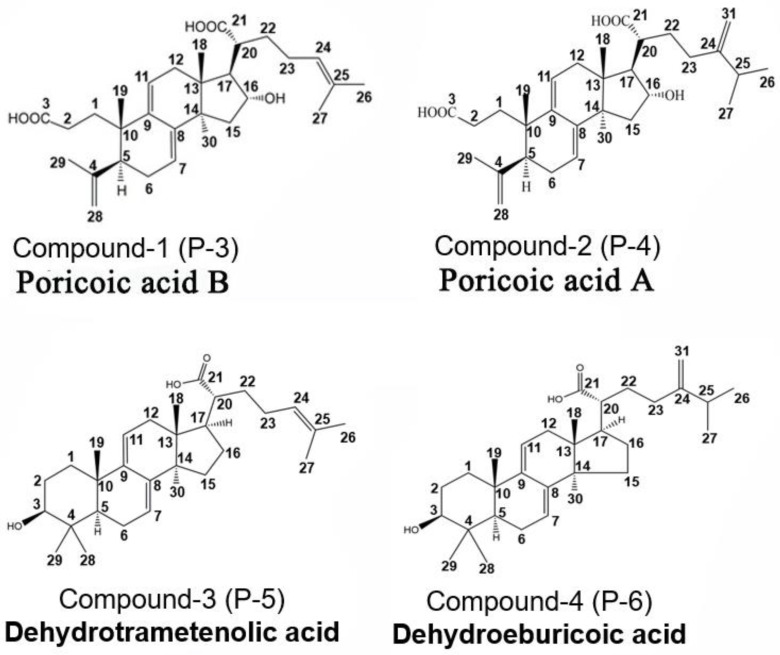
The structures of compounds **1**–**4**.

**Figure 4 foods-10-03155-f004:**
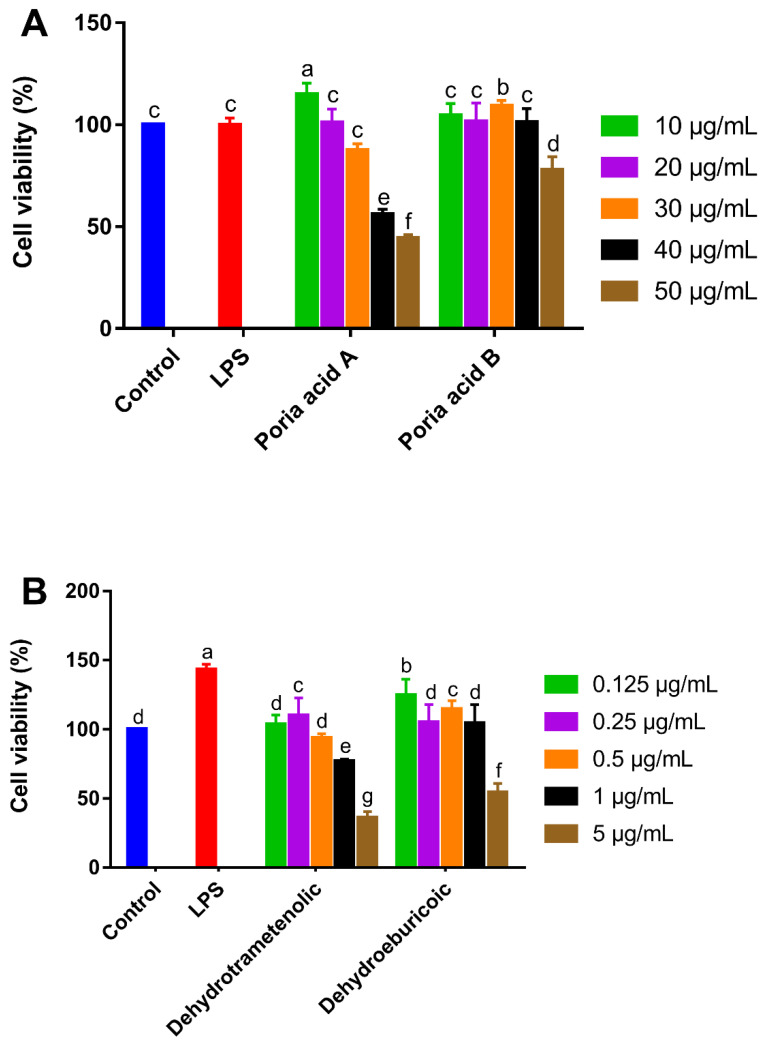
The effects of various concentrations of poricoic acid A and poricoic acid B (**A**), dehydrotrametenolic acid and dehydroeburicoic acid (**B**) on the cell viabilities of murine macrophage cells (RAW 264.7). The values are expressed as the mean ± SD (*n* = 3). Different letters (a–g) indicate significant differences (*p* < 0.05). LPS, lipopolysaccharide.

**Figure 5 foods-10-03155-f005:**
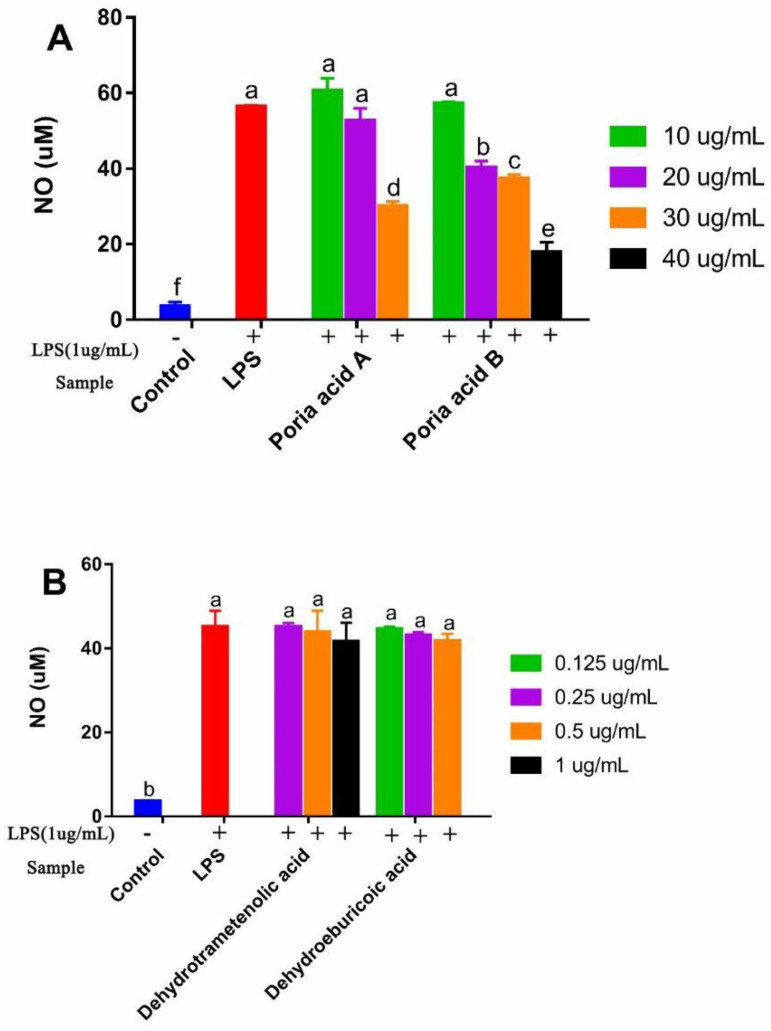
The effects of various concentrations of poricoic acid A and poricoic acid B (**A**), dehydrotrametenolic acid and dehydroeburicoic acid (**B**) on the NO production in lipopolysaccharide (LPS)-stimulated murine macrophage cells (RAW 264.7). The values are presented as the mean ± SD (*n* = 3). Different letters (a–f) indicate significant differences (*p* < 0.05).

**Figure 6 foods-10-03155-f006:**
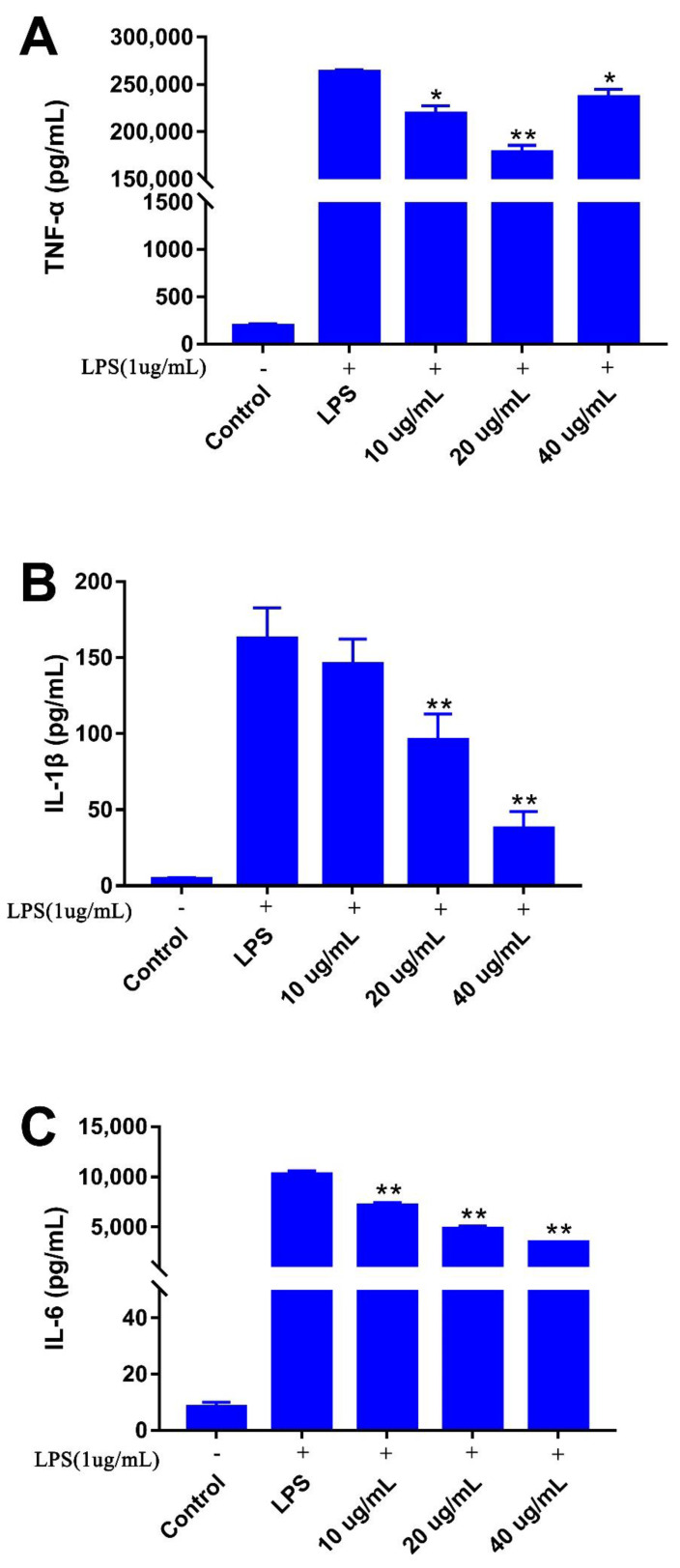
The effect of various concentrations of poricoic acid B on the productions of TNF-α (**A**), IL-1β (**B**), and IL-6 (**C**) in lipopolysaccharide (LPS)-stimulated murine macrophage cells (RAW 264.7). The values are presented as the mean ± SD (*n* = 3). * *p* < 0.05, ** *p* < 0.01 vs. LPS group.

## Data Availability

Not applicable.

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
