# Peer review of "Anti-Inflammatory Activity of Four Triterpenoids Isolated from Poriae Cutis"

_foods, 2021, doi:10.3390/foods10123155_

Round 1

Reviewer 1 Report

This paper is well written but only known triterpenoids were isolated. It is not necessary to provide NMR spectra of known molecules or they can be included in supplementary informations.

The interest of this article remains in the biological study because the terpenoids are not new. Thus, the authors have to discuss more their biological results : structure activity relationships, previous results with other terpenoids, etc...

Author Response

We are grateful for the detailed comments and suggestions provided by each of the reviewers, and we believe that their input has greatly improved our manuscript. You will find our general reply and point-by-point responds in the attachment to the reviewers’ comments/questions.

Reviewer 2 Report

Introduction: The authors should incorporate possible applications of Poria cutis in food and highlight reason for using in food.

Introduction in its current form is not sufficient, authors should provide more recent background information.

Methods: Poria cocos was purchased from local market, was there identification or confirmation was made???

The temperature used for extraction was 60 degrees, how authors confirmed that heat sensitive components were not lost?

Results: What was the recovery yield after extraction?

Conclusion is not appropriate, authors should not repeat results, instead concrete outcomes should be presented by emphasizing that objectives were met. Also include future applications.

The study overall has significant data, however the above mentioned suggestions can significantly improve the manuscript.  

Author Response

(The authors gave the same response as above.)

Reviewer 3 Report

Please place the known compounds NMR spectrum in the supplementary file. 

The 13C NMR spectrum is required to confirm the structure.

Please find other comments in the attached PDF file.

Author Response

(The authors gave the same response as above.)

Round 2

Reviewer 1 Report

This paper is now in a suitable form to be published 

Author Response

We are grateful for your comments and we believe that your input has greatly improved our manuscript.

Reviewer 2 Report

The manuscript has been revised as advised by the reviewers. 

Author Response

(The authors gave the same response as above.)
